# p53 Enhances *Artemisia annua* L. Polyphenols-Induced Cell Death Through Upregulation of p53-Dependent Targets and Cleavage of PARP1 and Lamin A/C in HCT116 Colorectal Cancer Cells

**DOI:** 10.3390/ijms21239315

**Published:** 2020-12-07

**Authors:** Eun Joo Jung, Won Sup Lee, Anjugam Paramanantham, Hye Jung Kim, Sung Chul Shin, Gon Sup Kim, Jin-Myung Jung, Chung Ho Ryu, Soon Chan Hong, Ky Hyun Chung, Choong Won Kim

**Affiliations:** 1Departments of Biochemistry, Institute of Health Sciences, Gyeongsang National University School of Medicine, Jinju 52727, Korea; eunjoojung@gnu.ac.kr (E.J.J.); cwkim@gnu.ac.kr (C.W.K.); 2Departments of Internal Medicine, Institute of Health Sciences, Gyeongsang National University Hospital, Gyeongsang National University School of Medicine, Jinju 52727, Korea; anju.udhay@gmail.com; 3Research Institute of Life Science, College of Veterinary Medicine, Gyeongsang National University, Jinju 52828, Korea; gonskim@gnu.ac.kr; 4Departments of Pharmacology, Institute of Health Sciences, Gyeongsang National University School of Medicine, Jinju 52727, Korea; hyejungkim@gnu.ac.kr; 5Department of Chemistry, Research Institute of Life Science, Gyeongsang National University, Jinju 52828, Korea; sshin@gnu.ac.kr; 6Departments of Neurosurgery, Institute of Health Sciences, Gyeongsang National University Hospital, Gyeongsang National University School of Medicine, Jinju 52727, Korea; gnuhjjm@gnu.ac.kr; 7Department of Food Technology, Research Institute of Life Science, Gyeongsang National University, Jinju 52828, Korea; ryu@gnu.ac.kr; 8Departments of Surgery, Institute of Health Sciences, Gyeongsang National University Hospital, Gyeongsang National University School of Medicine, Jinju 52727, Korea; hongsc@gnu.ac.kr; 9Departments of Urology, Institute of Health Sciences, Gyeongsang National University Hospital, Gyeongsang National University School of Medicine, Jinju 52727, Korea; kychung@gnu.ac.kr

**Keywords:** p53, *Artemisia annua* L., polyphenols, colorectal cancer, cell death, ROS, acidic vesicles

## Abstract

Plant-derived natural polyphenols exhibit anticancer activity without showing any noticeable toxicities to normal cells. The aim of this study was to investigate the role of p53 on the anticancer effect of polyphenols isolated from Korean *Artemisia annua* L. (pKAL) in HCT116 human colorectal cancer cells. We confirmed that pKAL induced reactive oxygen species (ROS) production, propidium iodide (PI) uptake, nuclear structure change, and acidic vesicles in a p53-independent manner in p53-null HCT116 cells through fluorescence microscopy analysis of DCF/PI-, DAPI-, and AO-stained cells. The pKAL-induced anticancer effects were found to be significantly higher in p53-wild HCT116 cells than in p53-null by hematoxylin staining, CCK-8 assay, Western blot, and flow cytometric analysis of annexin V/PI-stained cells. In addition, expression of ectopic p53 in p53-null cells was upregulated by pKAL in both the nucleus and cytoplasm, increasing pKAL-induced cell death. Moreover, Western bot analysis revealed that pKAL-induced cell death was associated with upregulation of p53-dependent targets such as p21, Bax and DR5 and cleavage of PARP1 and lamin A/C in p53-wild HCT116 cells, but not in p53-null. Taken together, these results indicate that p53 plays an important role in enhancing the anticancer effects of pKAL by upregulating p53 downstream targets and inducing intracellular cell death processes.

## 1. Introduction

Evidence suggests that there is an inverse relationship between the consumption of plant foods and the incidence of cancer [1,2]. Natural polyphenols in such foods are assumed to be responsible for the reduction of cancer incidence. Natural polyphenols can be classified as phenolic acids (e.g., caffeic acid), flavonoids (e.g., quercetin), polyphenolic amides (e.g., capsaicin) and other polyphenols (e.g., resveratrol, curcumin) [3]. Colorful plant foods contain plentiful natural polyphenols which harbor antioxidant, anti-inflammatory and anticancer activities. These polyphenols have been reported to safely generate anticancer effects without showing any noticeable toxicities [4,5].

Colorectal cancer is caused by the transformation of the normal epithelium into an invasive carcinoma through serial pathological changes. It is the third most common cancer in men worldwide, the second most common cancer in women, and the third leading cause of cancer mortality [3,6]. In Korea, the incidence of colorectal cancer is highest among elderly women [7]. The tumor suppressor p53, known as the “Guardian of genome”, still appears to be one of the major therapeutic targets for cancer due to its strong anticancer ability to respond to various cellular stress such as high reactive oxygen species (ROS), DNA damage, genomic instability, and senescence [8,9,10]. Inactivation of 53 signaling is a common feature of human cancer. Mutations in p53 are present in almost 50% of all colorectal cancers: the incidence of distal colon and rectal cancer is high; the incidence of proximal colon cancer is low [3,9]. Restoring or enhancing p53 function in cancer is considered as an effective and cancer-specific therapeutic strategy [11]. The p53 protein is upregulated by polyphenols from various dietary sources (e.g., apple polyphenol phloretin, grape polyphenol resveratrol), and activates anticancer signaling pathways [12,13,14]. In addition, natural polyphenols have been shown to induce apoptosis or other types of cancer cell death by activation of p53. Moreover, many cancers frequently exhibit loss of normal p53 function during the oncogenic process [3,9]. Thus, upregulated wild-type p53 by natural polyphenols in plant foods may be a main cause of the reduction in cancer incidence. However, natural polyphenols are known to induce anticancer effects even in p53-mutant cancer cells through inactivation of oncogenic cell signaling or cancer cell survival signaling pathways [15,16,17,18]. In addition, there are many instances where wild-type p53 has shown no effect or enhances cellular sensitivity to chemotherapy and radiation compared to mutant p53; wild-type p53 plays various role depending on cancer treatment [19,20,21]. Furthermore, to date, the role of p53 on the anticancer effects of natural polyphenols in various cancer cells is still unclear. Therefore, the understanding of the influence of an p53 expression on anticancer effects of phytochemical therapy from natural herbs or plant can help or trigger personalized medicine with the phytochemicals.

*Artemisia annua* L. is sweet wormwood plant, and its active ingredient, artemisinin, has been developed worldwide as an antimalarial drug [22]. The chemical structure of artemisinin is sesquiterpene lactone, and biological studies on *Artemisia annua* L. have been intensively carried out by artemisinin and its derivatives (dihydroartemisinin, artesunate, artemether, arteether) in various cancer cells [22,23]. Its derivatives also have anticancer activity [24,25,26], and its effect is affected by p53 [24]. In addition, the whole extract of *Artemisia annua* L. was shown to induce apoptosis by regulating the phosphorylation of PDK1 and Akt through the PTEN/p53-independent pathway in colon cancer cells [27]. However, little has been studied on the role of p53 in the anticancer effect of polyphenols isolated from *Artemisia annua* L. in colorectal cancer cells, especially in relation to the nuclear structure change and post-translational modifications of lamin A/C.

We previously characterized polyphenols isolated from Korean *Artemisia annua* L. (pKAL), demonstrating that pKAL inhibits adhesion, invasion, and EMT in highly metastatic breast cancer cells [28,29]. In this study, we compared the difference in cellular responses by pKAL between p53-wild and p53-null HCT116 human colorectal cancer cells to investigate an influence of endogenous p53 on pKAL-induced anticancer effects. In addition, we examined the role of ectopic p53 in pKAL-induced cell death signaling in p53-null HCT116 cells to prove the role of endogenous p53. Thus, here we show that p53 can promote pKAL-induced cell death by activating downstream targets and inducing intracellular processes associated with HCT116 cancer cell death.

## 2. Results

### 2.1. pKAL Induced ROS Production, PI Uptake, and Nuclear Structure Change in p53-Null HCT116 Colorectal Cancer Cells

To better understand the role of p53 in the anticancer effect of pKAL, we first assessed ROS production and dead cells caused by pKAL in p53-null HCT116 cells (HCT116-p53^−/−^) by fluorescence microscopy analysis of co-stained cells with DCF and propidium iodide (PI). As shown in Figure 1A, DCF-stained cells (green fluorescent) were significantly upregulated by pKAL treatment for 36 h (panels a,c), whereas only a few cells (cells 1,2 emitting red fluorescence) were stained by PI (panels c,d), suggesting that many high ROS generating cells are alive at 36 h after pKAL treatment. However, both green and red fluorescence emitting cells were greatly increased at 60 h by the increased pKAL treatment duration (Figure 1B). Among the green fluorescent cells, cells 1–3 stained by PI (dead cells) showed an assembly of fragmented nucleus, whereas cells 4–9 not stained by PI (live cells) emit peripheral cytoplasmic green fluorescence: indicating that cells are still alive even in high ROS production status (Figure 1B, amplified panels c’,d’). pKAL increased PI uptake and nuclear structure change in a dose-dependent manner: red fluorescence was more fragmented and scattered by pKAL treatment at 100 µg/mL than 50 µg/mL (Figure 1C). Nuclear structure change by pKAL were also confirmed by DAPI staining in a dose-dependent manner (Figure 1D). These finding show that pKAL can induce ROS production, PI uptake (cell death), and nuclear structure change even in the absence of endogenous p53 in p53-null HCT116 colorectal cancer cells, regardless of the correlation between ROS production and cell death.

### 2.2. pKAL Altered DNA Conformation and Acidic Vesicle Formation in p53-Null HCT116 Cells

We next examined the effect of pKAL on DNA conformational change and acidic vesicle formation by fluorescence microscopy analysis of acridine orange (AO) stained cells in p53-null HCT116 cells. AO was used as a nucleic acid-selective fluorescent cationic dye: green fluorescence by AO staining is primarily related to binding to DNA; red fluorescence by AO staining is associated with binding to RNA or acidic vesicles [30,31,32,33]. In this study, AO staining analysis revealed that strong green fluorescent cells were significantly increased by pKAL treatment for 60 h in a dose-dependent manner, representing large aggregate types or assembly of fragmented nuclear compartments in many cells (Figure 2A). Similarly, strong red fluorescent cells also increased significantly by 50 µg/mL of pKAL treatment for 60 h (Figure 2B). However, the expression pattern of red fluorescence was different from that of green fluorescence because it showed binding to RNA or acidic vesicles: the strong green fluorescence of cells 1 and 2 was considered as a nucleus with pKAL-induced DNA conformational change; in the same cells, strong red fluorescence was considered as acidic vesicles or lysosomes increased by pKAL in the cytoplasm; cells 3-6 showed strong round green fluorescence and a small portion of red fluorescence, without significant co-localization of green and red; however, most of the cells showed a co-localization pattern of green and red fluorescence; the green fluorescence of cells 7–13 was significantly co-localized with the red fluorescence in the same cell; especially, cells 9 and 13 exhibited a fragmented apoptosis nuclear phenotype (Figure 2B, amplified panels c’,d’). These findings show that pKAL induces DNA conformational change and acidic vesicles even in the absence of endogenous p53 in p53-null HCT116 colorectal cancer cells, supporting the results of Figure 1.

### 2.3. The Ability of pKAL to Suppress Cell Viability was Higher in p53-Wild HCT116 Cells than in p53-Null

In Figure 1 and Figure 2, pKAL showed anticancer effects associated with increased ROS level, PI uptake, nuclear conformation change, and acidic vesicle formation in 53-null HCT116 cells. To elucidate the potential role of endogenous p53 on the anticancer effect of pKAL, we first compared the differences in cell morphology and viability due to pKAL between p53-wild and p53-null HCT116 cells in hematoxylin staining and CCK-8 assay. As shown in Figure 3A, the results of hematoxylin staining showed that pKAL induced significant morphological change in both p53-wild and p53-null cells; however, small round cells were more induced by pKAL treatment in p53-null HCT116 cells than in p53-wild, especially at 50 µg/mL concentration (compare panels c,f). Moreover, the increase in small round cells by pKAL in p53-null cells resulted in a higher cell viability than in 53-wild, although a small round cell phenotype is usually found in dead cells (Figure 3B). These findings show that most of the pKAL-induced small round-shaped p53-null cells observed by hematoxylin staining are still alive, indicating that endogenous p53 may promote the anticancer effect of pKAL in p53-wild HCT116 cells.

### 2.4. The Ability of pKAL to Induce Apoptosis was Higher in p53-Wild HCT116 Cells than in p53-Null

To further elucidate the potential role of endogenous p53 in the anticancer effects of pKAL, we compared the differences in PARP1 cleavage, ROS production, and apoptosis caused by pKAL between p53-wild and p53-null HCT116 cells. Thus, Western blot analysis showed that endogenous p53 was upregulated by pKAL treatment for 48 h in p53-wild cells, but not in 53-null cells; the cleaved form of PARP1 was more induced by pKAL in a dose-dependent manner in p53-wild HCT116 cells than in 53-null (Figure 4A). However, flow cytometric analysis showed no significant difference in ROS production between p53-wild and p53-null cells treated with 50 µg/mL pKAL. In contrast, there was a significant difference in ROS production between the two cells treated with 100 µg/mL pKAL: higher ROS cells were observed in p53-null HCT 116 cells than in p53-wild; hypo-DCF-stained cells of 45.28% and hyper-DCF-stained cells of 11.44% in p53-wild cells; hypo-DCF-stained cell of 32.31% and hyper-DCF-stained cells of 32.46% in p53-null cells (Figure 4B). Moreover, flow cytometric analysis showed that both annexin V- and PI-stained cells were more increased by pKAL in p53-wild cells than in p53-null in a dose-dependent manner (Figure 4C). Taken together, these findings suggest that p53-null HCT116 cells can survive more than p53-wild cells under conditions of high ROS production induced by pKAL, and thus pKAL-induced apoptosis can be occurred more in p53-wild HCT116 cells than in p53-null, at least in part, through activation of endogenous 53 signaling.

### 2.5. Upregulation of Ectopic p53 by pKAL Resulted in the Increase of pKAL-Induced Nuclear Structure Change and Post-Translational Modification of Lamin A/C

To prove the role of p53 in the anticancer effect of pKAL, we first examined the effect of pKAL on the intracellular localization of ectopic p53 in p53-null HCT116 cells through immunofluorescence microscopy analysis. As shown in Figure 5A, green fluorescence for ectopic p53 was significantly upregulated by pKAL treatment for 48 h compared to the control (panel b vs. e,h,k). In the absence of pKAL treatment, ectopic p53 was mainly located in the nucleus, especially in the nuclear envelope (Figure 5A, panels a–c). However, ectopic p53 upregulated by pKAL was also somewhat located in the cytoplasm as well as in the nucleus (Figure 5A, panels d–i). In particular, significant nuclear deformation was associated with the upregulated expression of ectopic p53 by pKAL (Figure 5A, panels j–l). p53 is known to be upregulated by camptothecin (CPT), a DNA topoisomerase I inhibitor, and CPT has been shown to induce both p53-dependent and p53-independent cell death [34]. Consistent with previous results, Western blot analysis showed that ectopic p53 was upregulated by CPT, similar to pKAL treatment, resulting in the increase of cleavage and higher molecular weight modified form of lamin A/C (Figure 5B). These findings suggest that p53 can enhance pKAL-induced apoptotic cell death through activation of p53-dependent cell processing.

### 2.6. pKAL-Induced Cell Death was Enhanced by Ectopic p53 in p53-Null HCT116 Cells

To visualize pKAL-induced cell death enhanced by ectopic p53, we examined pKAL effect in p53-null HCT116 cells transfected with pCMV-p53 plasmid DNA via phase-contrast microscopy and trypan blue staining. Phase-contrast light microscopy analysis showed that cell morphology was significantly altered by both pKAL and CPT (Figure 6A, panels a–c), and the expression of ectopic p53 resulted in higher increase of fragmented dead cell phenotype in pKAL-induced morphological changes than CPT (Figure 6A, panels d–f). Consistent with these results, trypan blue-stained dead cells were significantly increased by both pKAL and CPT (Figure 6B, panels a–c), and the expression of ectopic p53 promoted more pKAL-induced cell death than CPT, resulting in more fragmented apoptotic phenotype (Figure 6B, panels d–f). These findings suggest that the additional activation system of ectopic p53 may further enhance pKAL-induced cell death than CPT in various p53-null cancer cells.

### 2.7. Upregulation of p53-Dependent Targets and Cleavage of PARP1 and Lamin A/C Were Associated with pKAL-Induced Cell Death Enhanced by p53

To elucidate the mechanisms involved in pKAL- and CPT-induced cell death enhanced by endogenous p53, we compared the effects of pKAL and CPT between p53-wild and p53-null HCT116 cells via Western blot analysis. As expected, endogenous p53 was upregulated by pKAL and more significantly by CPT in p53-wild cells, but not in p53-null (Figure 7A). The protein levels of p21, Bax and DR5, known as p53-dependent targets [35,36], were significantly upregulated by both pKAL and CPT in p53-wild cells, but not significantly regulated in p53-null; notably, DR5 was significant upregulated by pKAL in a concentration dependent manner, but not in the presence of co-treatment of CPT (Figure 7A). However, the CPT-induced upregulation of p53, p21 and Bax, not DR5, appeared to be downregulated by co-treatment of pKAL in a concentration dependent manner, which indicates that p53 signaling is not readily activated by pKAL treatment under conditions of DNA damage by CPT. Similarly, cleavage of caspase-8, PARP1 and lamin A/C was upregulated by pKAL and more significantly by CPT in p53-wild cells, but not in p53-null (Figure 7B). Caspase-8 cleavage by CPT was increased by pKAL, whereas PARP1 cleavage by CPT was rather decreased by pKAL in a concentration dependent manner (Figure 7B). However, the post-translationally modified high molecular form of lamin A/C was more dramatically upregulated in p53-null cells than in p53-wild, suggesting that these higher molecular weight form of lamin A/C may be involved in the inhibition of pKAL- and CPT-induced cell death (Figure 7B). Collectively, these finding show that p53 promotes pKAL-induced cell death through activation of p53-dependent targets and significant cleavage of PARP1 and lamin A/C in p53-wild HCT116 human colorectal cancer cells.

## 3. Discussion

Recent studies suggest that tumor suppressor p53 is still one of the strong therapeutic targets for cancer, and dietary polyphenols have a promising role to suppress tumor progression via activation of p53 signaling pathway [8,12,13]. The purpose of this research was to investigate the role of p53 on the anticancer effect of *Artemisia annua* L. polyphenols, named pKAL, in HCT116 human colorectal cancer cells. Although we previously demonstrated that pKAL induce anticancer effects on p53-mutant cancer cells, here we first revealed that pKAL exhibited anticancer effects in a p53-independent manner by inducing ROS production, PI uptake, nuclear structure change, and acidic vesicle formation in p53-null HCT116 cells (Figure 1 and Figure 2). In addition, studies on the role of endogenous and ectopic p53 have shown that p53 can enhance pKAL-induced cell death through activation of p53-dependent targets such as p21, Bax and DR5 and cleavage of PARP1 and lamin A/C (Figure 3, Figure 4, Figure 5, Figure 6 and Figure 7).

Lamin A and C are nuclear intermediate filament proteins that play a critical role in the nuclear structural maintenance, DNA replication, transcription and chromatin organization [37]. Proteolysis of Lamins is associated with collapsed nuclei and chromatin condensation in p53-dependent apoptosis [38]. Moreover, wild-type p53 stabilizes lamin A/C via direct interaction, and the stabilized lamin A/C with p53 binding (increased expression of lamin A/C) was associated with nuclear deformation [10]. These findings also support that p53 plays an important role in chromatin condensation, nuclear shrinkage and deformity by facilitating degradation of nuclear lamins [38,39], and induction of nuclear deformity by stabilizing lamin A/C which should be removed [10]. Therefore, the restoration or enhancement of p53 gene could be a good strategy for the treatment of pKAL in cancer patients who are not eligible for the conventional therapies due to the patient’s poor condition.

However, there are some results opposite to ours, suggesting that wild-type p53 may induce a drug resistance in the course of cancer therapy by inducing cell cycle arrest, in much weaker stimuli than required for cell death [40]. In addition, drug sensitivity of p53-wild type cells varies depending on the status of cell cycle, or p53 regulators [41]. In addition, there are many instances where wild-type p53 has shown no effect or enhances cellular sensitivity to chemotherapy and radiation compared to mutant p53 [19,20,21]. Hence, p53-mediated anticancer effects may vary depending on the anticancer drugs. To avoid this bias, we used anticancer drug, the natural compound camptothecin (CPT) found in the Chinese ornamental tree, Camptotheca acuminata. CPT is known to exhibit anticancer activity in various cancer cells [42,43,44]. In this study, we checked the effect of ectopic wild-type p53 on CPT-induced cell death in p53-null HCT116 cells. As shown in Figure 5B, CPT also induced the cleaved forms and higher molecular weight modified forms of lamin A/C, suggesting that the induction of higher molecular weight forms of lamin A/C is not an unique finding for pKAL-induced anticancer activity. In addition, ectopic expression of p53 increased the cleaved forms of lamin A/C and higher molecular weight modified forms of lamin A/C caused by CPT in p53-null HCT 116 cells (Figure 5B). Consistent with these results, ectopic wild-type p53 facilitated CPT-induced cell death, but not as much as pKAL-induced cell death (Figure 6B). These findings support that our results may be applicable to other natural phytochemical-induced cell death as well as pKAL-induced cell death. Moreover, although there are some variations in drug sensitivity of p53-wild type cells, many investigators still explain this by p53 dynamics (p53 expression levels and activation of p53 regulators) [41].

In this study, we traced the pKAL-induced cell fate by co-staining of DCF and PI. However, it was noted that pKAL-induced ROS production and PI uptake showed no significant causal relationship at 36 h after pKAL treatment in fluorescence microscopy analysis; pKAL-induced ROS were localized mainly in the cytoplasm in PI-unstained viable p53-null HCT116 cells, whereas ROS were localized mainly in the nucleus in PI-stained dead cells (Figure 1B, amplified panels c’,d’). These results suggest that p53-null HCT 116 cells can survive from high ROS production condition, and that the highly expressed cytoplasmic ROS are not directly involved in cell death. This finding could be supported by the fact that ROS play a dual role depending on the concentration of ROS, the amount of anti-oxidants, and the activated signaling pathways [45]. Cancer cells generally have higher basal ROS levels than their normal counterparts, and the elevated ROS promote tumor development and progression [46,47]. In addition, cancer cells express high levels of antioxidant proteins to protect cells from ROS-mediated damage [46,47]. However, disproportionally increased intracellular ROS can induce cancer cell cycle arrest, senescence, and apoptosis [46,47]. In this study, the proportion of cells with high ROS production during pKAL treatment was less in p53-wild HCT116 cells than in p53-null cells, whereas the proportion of dead cells was larger, suggesting that the intracellular ROS levels and signals could influence the fate of cancer cells, but not the high ROS production itself alone.

There are some limitations that we have not clearly elucidated in this study. The first one is that both ectopic p53 and lamin A/C were significantly post-translationally modified by pKAL in a certain mechanism (Figure 5B). Post-translational modifications of p53 and lamin A/C are known to phosphorylation, ubiquitination, sumoylation, acetylation, etc. [48,49]. Lamin A/C can also be cleaved into 47- and 37-kDa fragments during apoptosis, and lamin breakdown facilitates nuclear breakdown [38]. The role of the pKAL-induced high molecular weight modified forms of p53 and lamin A/C is currently not clearly elucidated, but the pKAL-induced cleaved forms of lamin A/C appear to be associated with nuclear deformation and fragmentation (Figure 5B). Similarly, p53 and Lamin A/C were also post-translationally modified by CPT, indicating that this phenomenon could be associated with nuclear breakdown or shrinkages, but needs further investigation (Figure 5B). The second one is that p53 upregulation by 100 µg/mL pKAL treatment was less than that by 50 µg/mL, indicating that the p53 protein may be degraded by a certain mechanism during a massive pKAL-induced cell death (Figure 4A). This finding can be explained by the fact that p53 can be poteasome-mediated degraded after ubiquitination by a number of E3 ubiquitin ligases, and that it also can be 20S poteasome-mediated cleaved following oxidative stress [50,51]. The third one is that little is still known about how p53 stabilizes the higher molecular weight modified forms and cleaved forms of lamin A/C (Figure 5B). Further investigation into this is required to explain this issue.

In conclusion, this study suggests that p53 can augment pKAL-induced anticancer effect by activating p53-dependent signaling and inducing cleavage of PARP1 and lamin A/C in HCT116 human colorectal cancer cells, and that the combination therapy of p53 and pKAL may be one of the efficient non-toxic chemotherapeutic strategies for cancer treatment. Future studies on other p53-dependent targets as well as p21, Bax and DR5 could contribute significantly to understanding the p53-dependent role on the anticancer effects of dietary polyphenols in various cancer cell types.

## 4. Materials and Methods

### 4.1. Reagents

Fetal bovine serum (FBS), penicillin/streptomycin, trypLE express with phenol red, and opti-MEM reduced serum medium were from Life Technologies (Carlsbad, CA, USA). RPMI 1640 medium was from HyClone (Logan, UT, USA). Cell counting kit-8 (CCK-8) was from Dojindo (Kumamoto, Japan). 2′,7′-dichlorofluorescein diacetate (DCF-DA), propidium iodide (PI), 4′6-diamidino-2-phenylindole (DAPI), and acridine orange (AO) were from Sigma (St. Louis, MO, USA). Annexin V-Fluos was from Roche (Mannheim, Germany). Hematoxylin solution was from Merck KGaA (Darmstadt, Germany). Trypan blue 0.4% solution was from Bioworld (Louis Park, MN, USA). FuGENE HD transfection reagent was from Promega (Madison, WI, USA). Aqueous mounting medium with anti-fading agents was from Biomeda (Foster, CA, USA). Acrylamide/bis-acrylamide 37.5:1 solution (40%), urea, Tween-20, and DMSO were from Amresco (Solon, OH, USA). Dishes, plates, tubes, and pipetts for cell culture were from SPL Life Sciences (Pocheon, Republic of Korea) or Thermo Scientific (Rockford, IL, USA). Amersham Protran 0.2 µm NC membrane was from GE Healthcare Life Sciences. ECL Ottimo Western blot detection kit was from TransLab (Daejeon, Republic of Korea). p53 (DO-1), PARP-1 (F-2), p21 (C-19), DR5 (N-19), Lamin A/C (E-1), and GAPDH (FL-335) antibodies were from Santa Cruz (Santa Cruz, CA, USA); β-actin (clone AC-15) antibody was from Sigma (St. Louis, MO, USA); Bax antibody was from BD Biosciences (San Jose, CA, USA); Caspase-8 (p18) antibody was from Invitrogen (Carlsbad, CA, USA). Secondary goat anti-rabbit and anti-mouse HRP conjugates were from Bio-Rad (Hercules, CA, USA) and Bethyl Laboratories (Montgomery, TX, USA).

### 4.2. pKAL Compounds

To isolate the pKAL compounds, mixed tissues including root, stem, leaf, and flower of *Artemisia annua* L. were lyophilized, ground, and extracted with 70% methanol at 60 °C for 20 h. The extract was filtered through a glass funnel and concentrated at 35 °C using a rotary evaporator. To remove the fat component, the concentrated aqueous extract was extracted three times with equal volumes of *n*-hexane and methylene chloride. The filtrate was extracted three times with ethyl acetate to isolate the pKAL compounds and dried over anhydrous magnesium sulfate. The pKAL compounds were identified by Prof. Sung Chul Shin using liquid chromatography-tandem mass spectrometry (LC/MS/MS) as previously described [28]. Quantification of polyphenol compounds was analyzed from peak areas of the HPLC-UV chromatograms at 280 nm using eight standards in quintuplicate measurements. Thus, the pKAL compounds were determined as caffeic acid, quercetin-3-*O*-galactoside, mearnsetin-glucoside, kaempferol-3-*O*-glucoside, quercetin-3-*O*-glucoside, mearnsetin-glucoside, ferulic acid, isorhamnetin-gluoside, diosmetin-7-*O*-d-glucoside, luteolin-7-*O*-glucoside, quercetin, quercetagetin-3-*O*-methyl ether, luteolin, 8-methoxy-kaempferol, quercetagetin-5,3-di-*O*-methyl ether, kaempferol, 3,5-dihydroxy-6,7,4′-trimethoxyflavone, 3,5-dihydroxy-6,7,3′,4′-tetramethoxyflavone, and isorhamnetin. Except for isorhamnetin, these pKAL compounds were identical to those identified in each tissue of the root, stem, leaf, and flower of *Artemisia annua* L. in previous studies [28]. For the experiment, pKAL compounds were dissolved in DMSO solvent at a concentration of 100 mg/mL and stored in a −20 °C freezer until use.

### 4.3. Cell Culture

p53-wild type HCT116 human colorectal cancer cells (HCT116-p53^+/+^) were purchased from Korean cell line bank (KCLB No. 20247). p53-null HCT116 (HCT116-p53^−/−^) colorectal cancer cells were originally provided by Dr. Bert Vogelstein (Johns Hopkins University School of Medicine, Baltimore, MD, USA). HCT116 and p53-null HCT116 cells were maintained in RPMI medium with L-glutamine (300 mg/L), 25 mM HEPES, 25 mM NaHCO_3_, 1% penicillin/streptomycin, and 10% heat inactivated FBS in a 37 °C incubator supplemented with 5% CO_2_ in a humidified atmosphere. For maintenance and some experiments, cells were split every 3 days on a culture dish.

### 4.4. Fluorescence Microscopy of DCF-Stained Cells

Measurement of DCF fluorescence is used to detect reactive oxygen species (ROS), which are chemically reactive molecules containing oxygen such as superoxide, hydrogen peroxide and hydroxyl free radicals [52]. Without removing medium, cells grown on a 6-well plate were stained with 5 µM DCF-DA for 20 min in a 37 °C CO_2_ incubator. DCF-stained green fluorescent cells were detected by fluorescence microscopy (Leica Microsystems).

### 4.5. Fluorescence Microscopy of PI-Stained Cells

PI cannot passively traverse into an intact plasma membrane of viable cells, but sequentially can pass through damaged or permeabilized membranes, and then binds to DNA [53]. Without removing medium, cells grown on a 6-well plate were stained with 1 µg/mL PI for 20 min in a 37 °C CO_2_ incubator. The resulting red fluorescent cells were detected by fluorescence microscopy (Leica Microsystems).

### 4.6. Fluorescence Microscopy of DAPI-Stained Cells

DAPI staining is used to visualize the nucleus in interphase cells and the chromosomes in mitotic cells [54]. Cells grown on a 6-well plate containing cover slips were washed with PBS, fixed with 4% formaldehyde solution, permeabilized with 0.5% Triton X-100/PBS at RT for 3 min, rinsed with H_2_O, and then incubated with 5 µg/mL DAPI/H_2_O at RT for 5 min. The cover slips were mounted on a slide glass using an aqueous mounting medium with anti-fading agents. DAPI-stained blue fluorescent cells were detected by fluorescence microscopy (Leica Microsystems).

### 4.7. Fluorescence Microscopy of AO-Stained Cells

Acridine orange (AO) is a cell-permeable dye that can cross the plasma membrane of viable cells, and AO staining is used to detect nucleus (DNA) by green fluorescence and also to detect autophagy and acidic vesicles such as lysosomes by red fluorescence [30,31,32,33]. Without removing medium, cells grown on a 6-well dish were stained with 1 µg/mL AO/PBS for 20 min in a 37 °C CO_2_ incubator. AO-stained green and red fluorescent cells were detected by fluorescence microscopy (Leica Microsystems).

### 4.8. Light Microscopy of Hematoxylin-Stained Cells

Hematoxylin (cationic) staining is used to detect nucleus (DNA, RNA and acid nucleoprotein) [55]. Cells grown on a 6-well plate were washed with PBS, fixed with 4% formaldehyde solution, washed with PBS and then stained with hematoxylin solution for 30 min at RT. The cells were washed with PBS and were analyzed in 90% glycerol/PBS solution by phase-contrast light microscopy in a 20× objective (Inf Plan Fluor 20× LWD, 0.45NA/7.1WD) (EVOS XL Core, Life Technologies) with 300× amplification.

### 4.9. Cell Viability Assay

Cells grown on a 24-well dish were incubated with maintenance medium containing 10% CCK-8 reagent for 1 h in a 37 °C CO_2_ incubator. The reaction solution (100 µL each) was then transferred to a 96-well dish and was analyzed by measuring the absorbance at OD_485 nm_ using a CHAMELEON microplate reader (Hidex).

### 4.10. Western Blot Analysis

Whole cells (attached and floating cells) were extracted with SDS sample buffer and were boiled for 5 min at 95 °C. The resultant proteins were separated using SDS-PAGE, and transferred to an NC membrane. The membrane was blocked for 30 min at RT in blocking buffer (3% skim milk, 0.1% Tween-20, PBS) and then incubated with primary antibody at 4°C overnight. The blot was then washed with PBST (0.1% Tween-20, PBS) three times for 10 min, and incubated with an HRP-conjugated secondary antibody in blocking buffer for 1–2 h. After being washed with PBST, the blot was analyzed with the ECL Western blot detection system.

### 4.11. Flow Cytometric Analysis of DCF- and Annexin V/PI-Stained Cells

Whole cells were collected, stained and analyzed by flow cytometry (FACS Calibur, Becton Dickinson). For DCF staining, cells were incubated with 5 µM DCF-DA for 20 min in a 37 °C water bath, and FL-1H green fluorescent cells were analyzed by flow cytometric analysis. For annexin V/PI staining, cells were incubated with annexin V-Fluos in 10 mM Hepes (pH 7.4)/140 mM NaCl/5 mM CaCl_2_/1 µg/mL PI/PBS solution for 20 min at RT, and FL1-H green and FL2-H red fluorescent cells were analyzed by flow cytometric analysis.

### 4.12. DNA Transfection

Plasmid DNA was incubated with FuGENE HD transfection reagent in opti-MEM reduced serum medium for 20–30 min at RT, and then transfected into p53 null HCT116 colorectal cancer cells. The next day, after replaced with fresh maintenance medium, the cells were treated with the drug for the indicated time.

### 4.13. Immunofluorescence Microscopy

Cells grown on a 6-well plate containing cover slips were washed with PBS, fixed with 4% formaldehyde solution, and permeabilized with 0.5% Triton X-100/PBS for 3 min at RT. After blocked with 1% BSA/PBS for 30 min, the cells were incubated with a primary antibody diluted in 1% BSA/PBS for 2 h, washed with PBS, and incubated with a secondary antibody for 1 h at RT. The cells were washed with PBS, rinsed with H_2_O, and then incubated with DAPI (5 µg/mL) in H_2_O for 5 min at RT. After rinsed with H_2_O, the cover slips were mounted on a slide glass using an aqueous mounting medium with anti-fading agents. Fluorescence images were obtained using fluorescence microscopy (Leica Microsystems).

### 4.14. Phase-Contrast Light Microscopy

Cell morphology was analyzed by phase-contrast light microscopy in a 10× objective (Inf Plan Achro 10X LWD PH, 0.25NA/6.9WD) (EVOS XL Core, Life Technologies) with 150× amplification.

### 4.15. Light Microscopy of Trypan Blue Stained-Cells

Trypan blue (negative charged) does not interact with the cell without a damaged membrane, and thus trypan blue staining is used to detect dead cells [55,56]. With the medium, the cells were treated with 0.1% trypan blue concentration for 20 min at RT, and were analyzed by phase-contrast light microscopy in a 20× objective (Inf Plan Fluor 20× LWD, 0.45NA/7.1WD) (EVOS XL Core, Life Technologies) with 300× amplification.

## Figures and Tables

**Figure 1 ijms-21-09315-f001:**
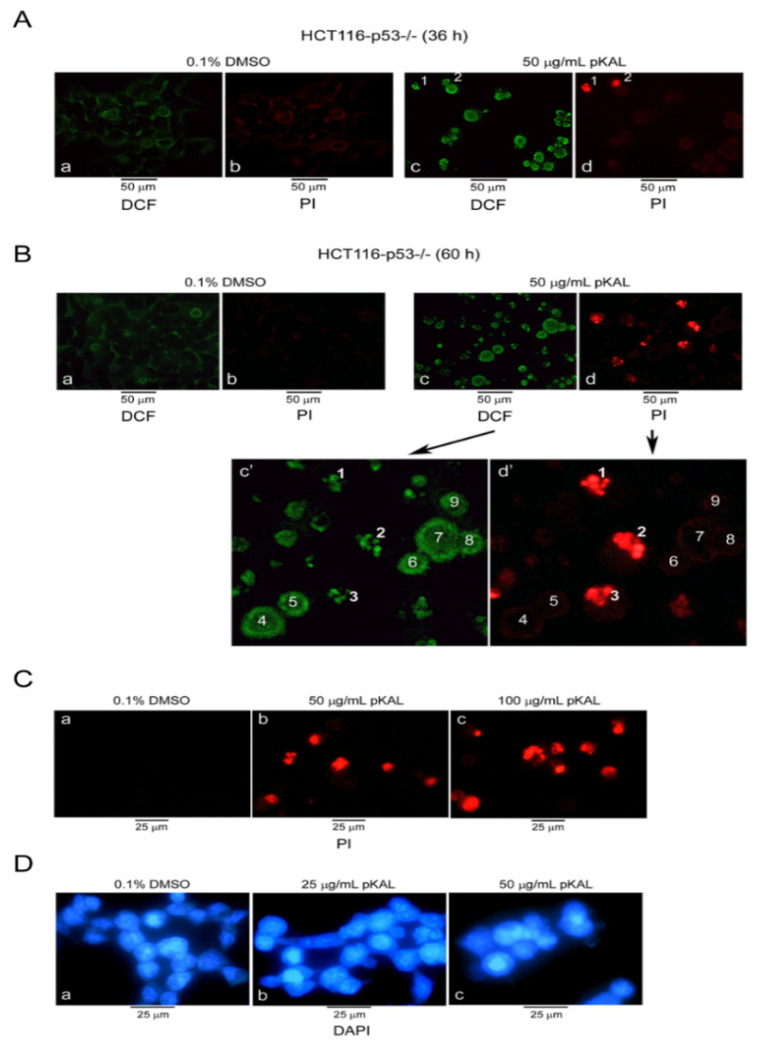
Effects of polyphenols isolated from Korean *Artemisia annua* L. (pKAL) on reactive oxygen species (ROS) production, propidium iodide (PI) uptake, and nuclear structure change in p53-null HCT116 cells. HCT116-p53^−/−^ cells were cultured with treatment of DMSO or pKAL at the indicated concentrations and times. (**A**,**B**) Cells were co-stained with 5 µM DCF-DA and 1 µg/mL PI for 20 min in a 37 °C CO_2_ incubator, and fluorescent images of green (DCF) and red (PI) were visualized by fluorescence microscopy. (**C**) PI staining. (**D**) DAPI staining.

**Figure 2 ijms-21-09315-f002:**
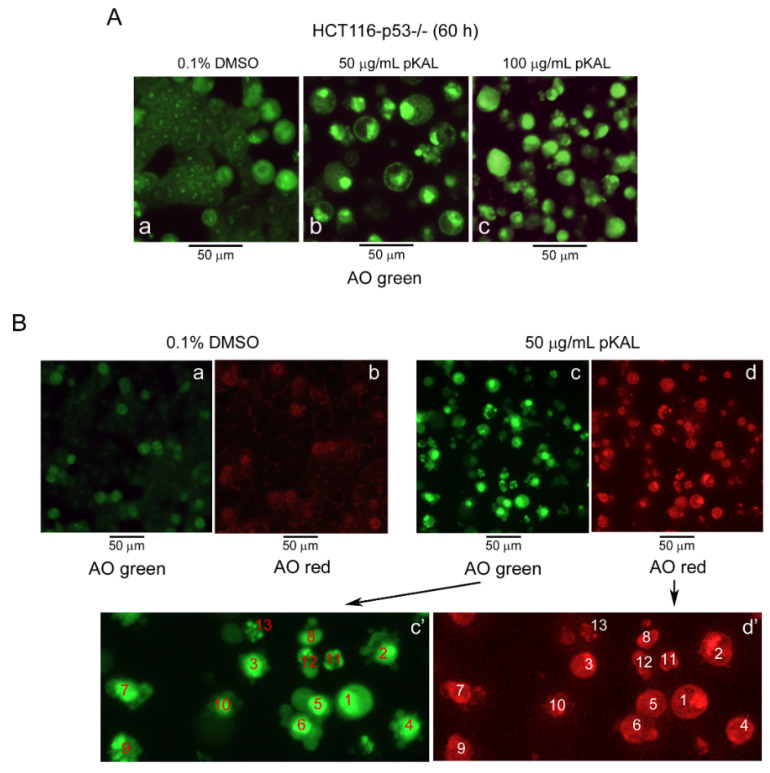
Effects of pKAL on DNA conformational change and acidic vesicle formation in p53-null HCT116 cells. HCT116-p53^−/−^ cells were grown for 60 h at the indicated concentration of DMSO or pKAL. (**A**,**B**) Cells were stained with 1 µg/mL acridine orange (AO), and fluorescent images for green and red were visualized by fluorescence microscopy.

**Figure 3 ijms-21-09315-f003:**
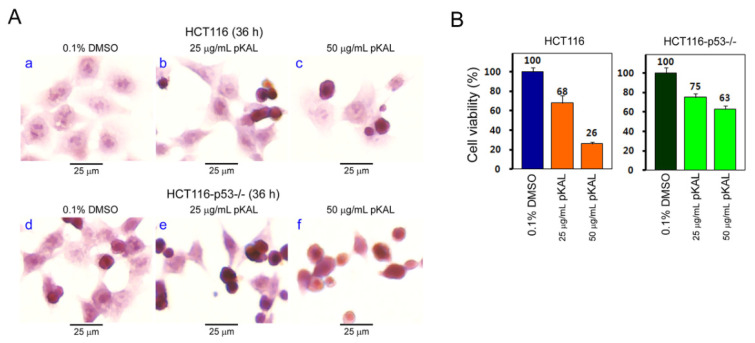
Comparison of pKAL effects on cell morphology and viability between p53-wild and 53-null HCT116 cells. HCT116 and HCT116-p53^−/−^ cells were grown for 36 h at the indicated concentration of DMSO or pKAL. (**A**) Light microscopy of hematoxylin-stained cells. (**B**) Cell viability was determined with a CCK-8 assay. Each bar represents the mean ± SD of three experiments.

**Figure 4 ijms-21-09315-f004:**
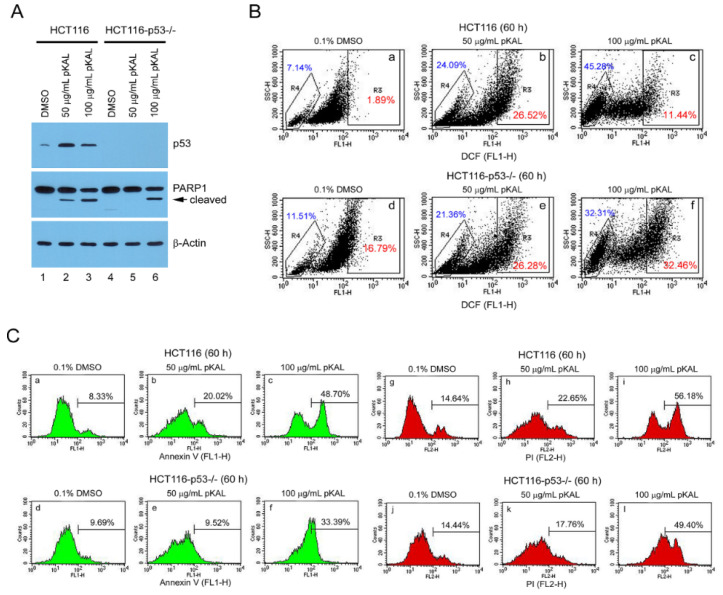
Comparison of pKAL effects on PARP1 cleavage, ROS production, and apoptosis between p53-wild and p53-null HCT116 cells. (**A**) HCT116 and HCT116-p53^−/−^ cells were grown for 12 h, and then treated with the indicated concentrations of DMSO or pKAL for 48 h. Cells were analyzed by Western blot using p53, PARP1 and β-actin antibodies. (**B**,**C**) HCT116 and HCT116-p53^−/−^ cells were grown for 60 h at the indicated concentration of DMSO or pKAL: (**B**) Flow cytometric analysis of DCF-stained cells; (**C**) Flow cytometric analysis of annexin V/PI-stained cells.

**Figure 5 ijms-21-09315-f005:**
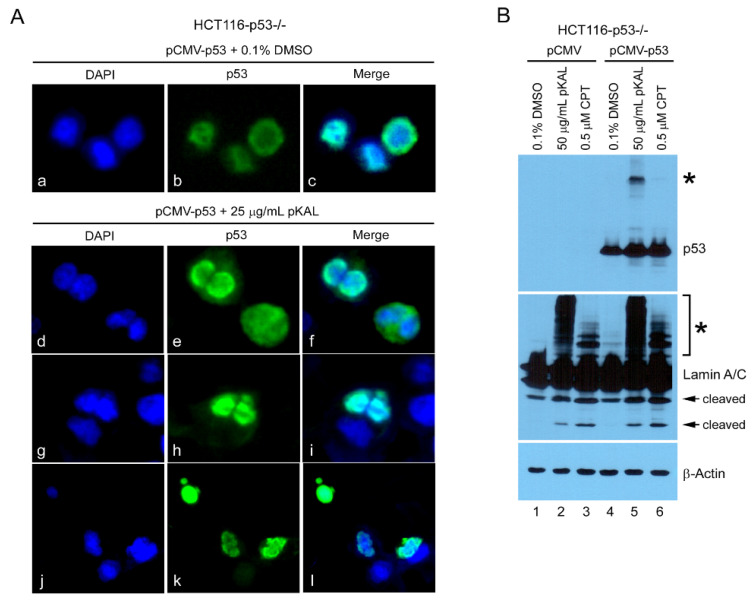
Effect of ectopic p53 upregulated by pKAL in nuclear structure change and post-translational modification of lamin A/C in p53-null HCT116 cells. (**A**) HCT116-p53^−/−^ cells were transfected with 1 µg pCMV-p53 and then treated with 0.1% DMSO or 25 µg/mL pKAL for 48 h. Cells were analyzed by immunofluorescence microscopy; DAPI (blue), p53 (green), and Merge (DAPI and p53). (**B**) HCT116-p53^−/−^ cells were transfected with 2 µg of pCMV or pCMV-p53 plasmid DNA and then treated with 0.1% DMSO, 50 µg/mL pKAL or 0.5 µM CPT for 24 h. Cells were analyzed by Western blot using p53, lamin A/C and β-actin antibodies. The asterisks indicate the higher molecular weight post-translational modified forms of p53 and lamin A/C, respectively, whereas the arrows indicate the lower molecular weight cleaved forms of lamin A/C.

**Figure 6 ijms-21-09315-f006:**
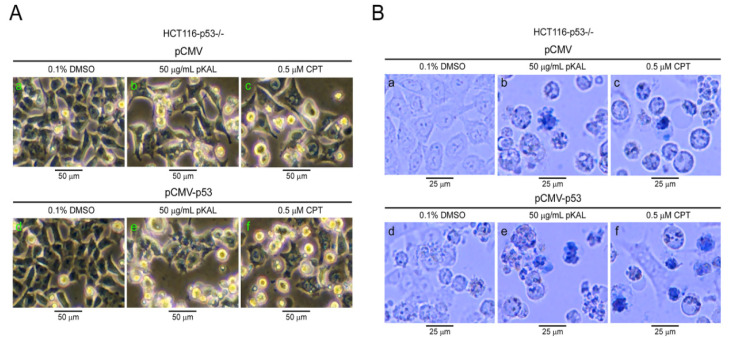
Effect of ectopic p53 on pKAL-induced morphological change and cell death in p53-null HCT116 cells. HCT116-p53^−/−^ cells were transfected with 1 µg of pCMV or pCMV-p53 plasmid DNA and then treated with 0.1% DMSO, 50 µg/mL pKAL or 0.5 µM CPT for 48 h. (**A**) Phase-contrast light microscopy. (**B**) Phase-contrast light microscopy of trypan blue-stained cells.

**Figure 7 ijms-21-09315-f007:**
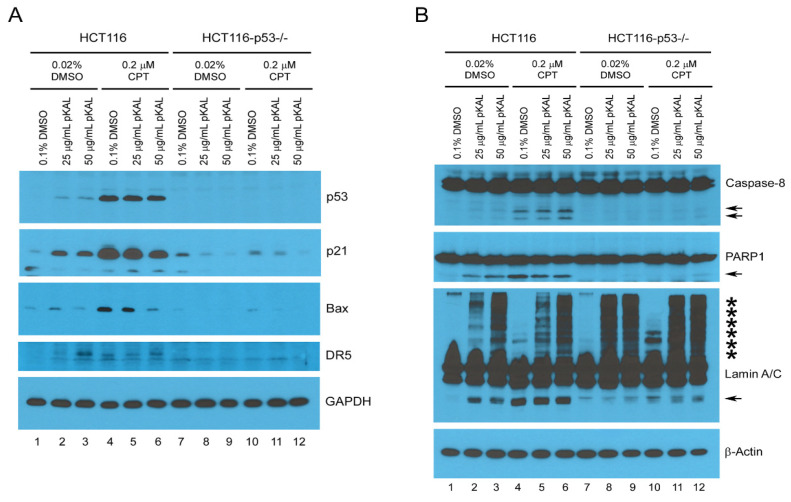
Comparison of cell death mechanisms by pKAL and CPT between p53-wild and p53-null HCT116 cells. (**A**,**B**) HCT116 and HCT116-p53^−/−^ cells were split on a 10 cm culture dish the day before, and then treated with the indicated amounts of drugs for 40 h. The cells were analyzed by Western blot using the indicated antibodies; the arrows indicate the lower molecular weight cleaved form(s) of the corresponding protein, and the asterisks indicate the higher molecular weight post-translational modified forms of lamin A/C (**B**).

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
