# Peer review of "p53 Enhances Artemisia annua L. Polyphenols-Induced Cell Death Through Upregulation of p53-Dependent Targets and Cleavage of PARP1 and Lamin A/C in HCT116 Colorectal Cancer Cells"

_ijms, 2020, doi:10.3390/ijms21239315_

Round 1
Reviewer 1 Report
Article
p53 enhances Artemisia annua L. polyphenols-induced cell death through upregulation of p53-dependent targets and cleavage of PARP1 and lamin A/C in HCT116 colorectal cancer cells
A brief summary
The title fits the presented problems. It is really well thought out and logically constructed paper, which based on a reliable and deep query presents the current state of knowledge. Work is really good constructed, however some corrections are required.
Any research explaining the effects of compounds isolated from plants is very valuable and highly appreciated. However, in order to fully assess the effects of plant-derived natural compounds, it is necessary to precisely describe plant material and how it was prepared for the research.
Specific comments
Line 44 – In my opinion keywords should not contain phrases used in the title. Instead of “p53”, “Artemisia annua L. polyphenols”, “pKAL”, “cell death”, “p53-dependent targets”, “cleavage of PARP1 and lamin A/C”, “53-null HCT116” and “colorectal cancer” I propose: “ROS”, “PI uptake”, “nuclear structure”, “DNA conformation”, “acidic vescile”, “cell morphology” and “cell vitability”; “anticancer effect” is OK.
Line 31, 90 – What does it mean “Korean Artemisia annua L.” – is it harvested from its natural state in Korea or grown on Korea commercially? Is it a local variety or chemotype of this species? Has anyone confirmed the identity of this species?
Line 391 – The sentence requires some clarification. It seems to me that you cannot isolate pKAL directly from the roots and other plant parts with LC/MS/MS.
1. From which part of the plant were phenols isolated – root, steam, leaf, flower or all mixed together? In the case of medicinal plants, usually the part of the plant is used that contains the most active compounds (as the Authors themselves describe [28]) – the herbal raw material.
2. Were polyphenols extracted from fresh, frozen or dried raw material? How was the raw material dried (methods, temperature) and crushed?
3. What solvent and how were the compounds extracted (Soxhlet apparatus, under reflux, sonication bath, ASE, ect.)?
4. How was the extract prepared for research, how was it concentrated / diluted?
5. What was the content of each identified compound?
Reviewer 2 Report
The work entitled "p53 enhances Artemisia annua L. polyphenols- induced cell death through upregulation of p53- dependent targets and cleavage of PARP1 and lamin A / C in HCT116 colorectal cancer cells" is a piece of decent paper. The authors Euo Joo Jung et al. Presented studies on p53 in the antitumor effect of polyphenols isolated from the Korean Artemisia annua L. (pKAL) in human HCT116 colon cancer cells. The work is very interesting and presents a multifaceted approach to the subject.
The publication is acceptable in my opinion and requires only a few corrections, which are rather a minor and insignificant correction than take anything away from the quality of this publication.
My comments are:
1. The keywords section should be improved, it is probably too detailed for the time being. Potential readers will probably find it hard to track this topic online, and that's not what the authors mean.
2. Since the work concerns polyphenols, it would probably be appropriate to quote authors such as A. Oniszczuk et al. e.g. Opuntia Fruits as Food Enriching Ingredient, the First Step towards New Functional Food Products, to which most of the work is concerned. Or: Oniszczuk A.et al: Content of phenolic compounds and antioxidant activity of new gluten-free pasta with addition of chestnut flour. Does Starzak et al. e.g. Anti-hypochlorite, antioxidant, and catalytic activity of three polyphenol-rich super-foods investigated with use of coumarin-based sensor. Is Influence of in vitro Digestion on Composition, Bioaccessibility and Antioxidant Activity of Food Polyphenols - Non-Systematic Review. K. A. Wojtunik-Kulesza et al. There are many other works available on the web. I have mentioned only the most familiar to me. But the authors do not have to be guided by this, although they may.
3. You surely need to correct the captions under the drawings, they are terribly not clear, unless only in my copy? Some are even tragic, you can't see anything on them.
4. Discussion section - needs to be significantly developed for now is a slight duplication of the results section.
5. There is no summary section - one sentence is definitely not enough for a magazine of this category - it needs to be corrected. And the presented conclusion by the authors is really not commented on.
At the very beginning, the work is great to read, but the further, the worse. It's a pity. I was counting on much more, the work needs to be improved if it is to be published. You can't.
I recommend major revision.
Round 2
Reviewer 2 Report
The authors have responded to my comments to a satisfactory degree.
A very good and interesting publication.
I recommend publishing the work.